# *COMT* Val/Met and Psychopathic Traits in Children and Adolescents: A Systematic Review and New Evidence of a Developmental Trajectory toward Psychopathy

**DOI:** 10.3390/ijms23031782

**Published:** 2022-02-04

**Authors:** Tuana Kant, Emiko Koyama, Clement C. Zai, Joseph H. Beitchman, James L. Kennedy

**Affiliations:** 1Institute of Medical Science, University of Toronto, Toronto, ON M5S 1A1, Canada; clement.zai@camh.ca (C.C.Z.); joe.beitchman@camh.ca (J.H.B.); 2Brain Science Department, Campbell Family Mental Health Research Institute, Centre for Addiction and Mental Health, Toronto, ON M5T 1R8, Canada; emiko.koyama@mail.utoronto.ca; 3Margaret and Wallace McCain Centre for Child, Youth and Family Mental Health, Centre for Addiction and Mental Health, Toronto, ON M6J 1H4, Canada; 4Department of Psychiatry, University of Toronto, Toronto, ON M5S 1A1, Canada; 5Laboratory Medicine and Pathobiology, University of Toronto, Toronto, ON M5S 1A1, Canada

**Keywords:** youth psychopathic traits, catechol-O-methyltransferase, COMT, Val158Met, genetic biomarkers, aggression, conduct disorder, oppositional defiant disorder, callous-unemotional traits

## Abstract

Psychopathic traits in youth may lead to adult criminal behaviors/psychopathy. The Val158Met polymorphism of catechol-O-methyltransferase (*COMT*) may influence the risk for psychopathy-related behaviors, while acting as a biomarker for predicting treatment response to dopaminergic medications. The literature shows inconsistent findings, making the interpretation of *COMT*’s role difficult. The aims of this article are (i) to conduct a systematic review to analyze the effects of *COMT* Val158Met on psychopathic traits in children and adolescents, and (ii) to present new evidence on the developmental trajectory of the association of Val158Met and youth psychopathic traits. For the systematic review, a literature search was conducted using PubMed, EMBASE, OVID Medline and PsychINFO with the search terms for psychopathic traits, Val158Met and age of interest. In our genotype study, the *COMT* Val158Met genotype of 293 youth with European ancestry was analyzed in association with the psychopathy-related behavior scores from the Child Behavior Checklist and the Psychopathy Screening Device. To examine the potential influence of developmental changes, the sample was split into at or above and below age 13, and analyses were performed in males and females separately. The literature search yielded twenty-eight articles to be included in the systematic review, which demonstrated mixed results on the association depending on environmental factors, sex ratios, age groups and behavioral disorder diagnoses. The results from our genotype study revealed that Met homozygous youth in the below age 13 group and conversely Val carrier youth in the above age 13 group were more likely to display psychopathic traits. To our knowledge, this is the first study to systematically review the effects of *COMT* Val158Met on psychopathic traits in childhood and adolescence, and to provide new evidence on the changing effects of Val158Met on psychopathy-related behaviors with development. Elucidating the role of the *COMT* genotype in conjunction with the child versus adolescent stage of development for psychopathic traits may help predict treatment response, and may lead to early intervention and prevention strategies.

## 1. Introduction

### 1.1. Psychopathy

Psychopathy is a neurocognitive personality disorder that involves emotional, interpersonal and behavioral symptoms such as lack of empathy, reduced guilt, increased criminal and violent activity, increased narcissistic tendencies and increased antisociality [1,2]. Adults with psychopathy tend to exhibit increased aggressive behaviors when compared to the rest of the population, and specifically increased proactive and purposeful aggression when compared to adults with other behavioral disorders. With severe impulsive and externalizing behaviors, psychopathy can have a range of symptoms across individuals. Blair et al. [1] suggested that the ultimate causes of psychopathy are genes, physical and sexual abuse, and neurobiological dysfunction from brain damage. While there is extensive research for neurobiology, genetics and environmental factors underlying psychopathy, results are inconclusive [2]. Within a given individual, psychopathic traits have high stability over time [3,4,5]. However, the developmental trajectory of psychopathy and its phenotypic expression can be influenced by different factors including genetic predispositions, neural connectivity and environmental factors [6]. Most would agree that in order to understand the underlying mechanisms of psychopathy in adults, it is crucial to understand the development of the disorder from early life. 

While the construct of psychopathy is used to define a personality disorder in adults, previous literature has used the term psychopathic traits to describe behavioral tendencies that youth may express at early stages of life [5,6,7,8,9,10]. There are still disagreements in extending the construct of psychopathy into children and adolescents due to reasons such as plasticity of personality during development, the potential stigmatization that the label may bring, as well as potential implications in the legal context [6]. However, research suggests that extending the concept of psychopathy to youth may provide the necessary foundation to detect the potential risk of exhibiting adult psychopathy from early behaviors in children and adolescents [7,8]. Therefore, similar to the aforementioned previous literature on youth, the rest of the manuscript will use the terms psychopathic traits and psychopathy-related behaviors to explain certain behaviors in youth that have been shown to resemble adult psychopathy and may be precursors to psychopathy in adulthood.

### 1.2. Psychopathic Traits in Youth

Psychopathic traits start early in life and have a high rate of stability across the development from childhood to adolescence [5]. These traits can be expressed differently among individuals and may include personality styles and antisocial behaviors such as rule breaking, impulsivity, narcissism, egocentrism, risk taking and extensive and persistent aggression [6,11]. While there is no set definition for measuring psychopathy, researchers have shown that there are different sub-categories for defining psychopathy-related behaviors in youth. 

The most studied idea to recognize and conceptualize psychopathic traits in children is by looking at the callous-unemotional (CU) traits in conduct disorder (CD) [12,13]. CU traits identify a group of children who have low empathy, lack of guilt, interpersonal callousness, restricted affect and a lack of concern for performance [14]. CD starts in childhood, where youth exhibit high aggression and rule-violating behaviors [11], along with an increased risk for adult psychopathy. Meanwhile, the risk of antisocial behavior in adulthood is increased in youth exhibiting CU traits along with CD [14,15].

In addition to CD and the CD-CU subcategory, psychopathy-related behaviors can also be found in children with other behavioral disorders. The early signs of CD can be found in children with attention deficit hyperactivity disorder (ADHD) [11,16] and oppositional defiant disorder (ODD) [11,16]. Children with ADHD usually exhibit symptoms of inattention, hyperactivity and impulsivity, whereas children with ODD usually exhibit symptoms of excessive aggression, impulsivity and rule breaking [11]. ADHD is highly comorbid with conduct problems [17]. Accordingly, being diagnosed with ADHD, ODD or CD increases the chances of forming CU traits and behavioral problems in childhood and adolescence [11,16].

Although children with a behavioral disorder, such as ODD or CD, combined with CU traits have a significantly worse prognosis and response to treatment, children with a behavioral disorder without CU traits are not free from the risk of psychopathy in adulthood [6]. The literature suggests that focusing solely on CU traits and this subtype of CD while conceptualizing psychopathic traits in children may not be sufficient to detect a heterogeneous group of risk subjects [6,18]. Therefore, it has been suggested that including the combination of interpersonal grandiose-manipulative (narcissism, manipulative and egocentric, superficial charm) and behavioral daring-impulsive (impulsivity, novelty seeking, antisocial and irresponsible behaviors) traits along with the affective CU traits may provide a more comprehensive diagnostic measure for psychopathic traits in youth [6,18,19]. 

As these early expressions of the psychopathic traits and behavioral problems in youth can lead to psychopathy and antisocial personality disorder (ASPD) in adulthood [11,20], understanding and detecting the risk factors may help in prevention and early intervention.

### 1.3. Psychopathic Traits and Genetics

Genetic factors have been associated with both the risk for exhibiting putative precursors of adult psychopathy and the stability during development [21,22,23], with 50% heritability [24]. Variations in different target genes in the dopaminergic pathway have been shown to contribute to the risk of psychopathy in adults and psychopathy-related behaviors in youth [25,26,27,28]. One of the target genes that have been highly studied is the catechol-O-methyltransferase (*COMT*) gene.

#### 1.3.1. Catechol-O-Methyltransferase (COMT)

The *COMT* gene is a dopaminergic gene that is located on chromosome 22q11.2 and encodes for the COMT enzyme [29] COMT breaks down active catecholamines such as dopamine, epinephrine and norepinephrine [30] to their inactive forms, thus controlling the levels of extracellular catecholamines in the prefrontal cortex (PFC) [31,32]. As the PFC is a crucial region for personality, cognition and executive function, the function and the activity levels of the COMT enzyme could play an important role in controlling these domains [33]. 

The most commonly studied single-nucleotide polymorphism (SNP) that may modify the activity level of the COMT enzyme is Val158Met (rs4680). This non-synonymous SNP changes the G nucleotide to an A, resulting in a change in the 158th amino acid from valine (Val) to methionine (Met). Met/Met carriers have been shown to have 3–4-fold reduction in COMT activity when compared with the Val/Val carriers [34,35]. Therefore, the Met allele results in less degradation, thus an increase in the dopamine and adrenaline levels in the PFC when compared with the Val allele [34].

#### 1.3.2. Psychopathic Traits and COMT

It is hypothesized that the catecholamine pathways and catecholamine levels may influence and alter the formation and expression of psychopathic traits [10,36,37]. As COMT regulates the levels of these neurotransmitters, it may be a good target gene for understanding the genetic etiology of psychopathic traits. Animal studies have shown that *COMT*-deficient mice and mice with low enzyme activity of COMT are associated with increased aggression in males [38]. Agreeing with the evidence from animal studies, studies with humans have also reported that children with one copy of the Met allele had an increased risk for aggressive behaviors [39]. However, there are inconsistent findings in the literature, where some studies demonstrated an association between the Val allele and increased externalizing behavior problems [40], while others demonstrated no association between the polymorphism and the oppositional defiant and conduct problems [41], thus making the interpretation of Val158Met’s role difficult. 

There are different factors that may affect the interaction between genes and psychopathic traits, thus leading to the inconsistent findings. Firstly, the environment has been demonstrated to have a significant role in the association, where these experiences may act as an increased risk factor or a buffer against the risk gene variant [42,43]. Another variable that may be important to consider is sex, as behavioral problems may manifest differently between males and females, and the *COMT* polymorphism may have different effects in males than in females [44,45]. Behavioral disorder diagnoses may also change the pathway through which the gene affects the psychopathic traits, such that a gene variant may be a risk factor for antisocial behavior in youth with ADHD, while having no effect on behavior in the general population [46].

Developmental age is another factor that may influence the association between Val158Met and psychopathic traits in youth. The activity levels of COMT increase through development [47], thus pointing to a possible change in the effects of *COMT*. Previous literature demonstrated that the influence of Val158Met may change through development, where Val158Met may have different effects on adolescents’ versus adults’ delay-discounting behavior [48] and in resting state functional brain connectivity [49]. Moreover, the results from our previous study, demonstrating that the effects of *MAOA*-uVNTR on psychopathic traits may change through development, further point to a differential effect that the gene variants may have in children versus in adolescents [27]. Similarly, research has previously demonstrated that the functional role of Val158Met in the pre-frontal cortex may change through development, where the adult pattern of the effects of Val158Met starts to appear after age 12, with the start of puberty and of changing dopaminergic pathways [50]. On the other hand, development from childhood to adolescence may also lead to both physiological and phenotypical changes in youth. While the interaction between the dopaminergic pathways and genes may change through development from childhood to adolescence [51], the frequency and expression of psychopathy-related behaviors may also change starting from grade seven, or approximately age 13 [11,52]. Therefore, consistent with previous literature [27,47,53], age 13 may represent a division point from childhood to adolescence to study the effects of Val158Met during development. The changing effects of Val158Met as well as the physiological and phenotypical changes in youth through development may explain the inconsistent findings in the literature so far.

These demonstrate that while analyzing the literature for Val158Met’s role in the early stages of psychopathic traits, it is crucial to consider and recognize environmental factors, sex, behavioral disorder diagnoses and developmental age as they may influence the gene–behavior association, leading to the inconsistent findings in the literature and making the interpretation of *COMT* Val158Met’s overall role challenging. 

Other *COMT* SNPs have also been studied in association with psychopathic traits and related behavioral problems. While children heterozygous for rs6269 (at the promoter region) and rs4818 (Leu136Leu) demonstrated increased aggressive behaviors and callous-unemotional traits when compared to homozygous children [54], rs4633 (His62His) did not show a significant association [54,55]. Moreover, adolescents with the GG genotype of rs6267 (Ala72Ser) demonstrated increased risk of aggressive behaviors when compared to T allele carriers [56]. However, while Val158Met has been studied thoroughly in different populations with different phenotypes, there are limited studies on the association between other *COMT* polymorphisms and psychopathic traits, thus requiring more research to make a conclusion regarding their effects.

There are diverse antipsychotic and stimulant drugs that influence the dopaminergic pathways, which have been used in the treatment of aggression and externalizing behaviors in youth, including aripiprazole and risperidone [11]. However, the regulation and expression of *COMT* Val158 versus Met158 may affect treatment response to dopamine system medications. Previous studies have demonstrated that adults showed genotype-specific treatment responses to the COMT-inhibiting drug tolcapone depending on their Val158Met genotype. Tolcapone improved executive functioning in the Val/Val group, and decreased it in the Met/Met group [57]. The results indicate that the Val158Met genotype may be used as a biomarker in developing personalized medication and in predicting treatment response to drugs for psychopathic traits. However, the possible role of Val158Met in treatment response has not been studied extensively in youth. 

Previous review articles have focused on the association of different candidate genes [58], and there has been limited focus in the literature on *COMT* [59], with no review on the role of *COMT* on psychopathic traits focusing on youth. Moreover, the changing effects of certain genes on psychopathic traits from childhood to adolescence [27], a change in the effects of *COMT* on ADHD from early childhood to middle childhood [60], and the dynamic nature of the *COMT* gene’s contribution to executive functioning from childhood to adolescence [50] point to a differential effect of Val158Met on psychopathic traits during development. While the changing effects of Val158Met have been demonstrated on different phenotypes [48,49], no research has been performed on the effects of Val158Met on psychopathic traits through development from childhood to adolescence. The aims of this paper are to conduct a systematic review of the literature and to present new data on the effects of Val158Met on psychopathic traits separately in children and adolescents, while emphasizing the possible role of Val158Met in the prediction of treatment response for psychopathic traits in youth.

## 2. Materials and Methods

### 2.1. Systematic Review—Data Selection

The current systematic review of the literature was conducted following the Preferred Reporting Items for Systematic reviews and Meta-Analyses (PRISMA) [61] guidelines. To analyze the *COMT*’s effects on psychopathic traits in children and adolescents, a literature search was conducted with the key words for psychopathic traits and endophenotypes of psychopathy (narcissis*, OR aggressi*, OR violen*, OR risk-tak* OR crim* OR antisocial OR ASB*, OR ASPD, OR APD, OR psychopath*, OR CU, OR “Callous-Unemotional Traits”, OR “Conduct Disorder”, OR “Conduct Problems”, OR CD, OR impulsiv*, OR ODD, OR “oppositional defiant disorder”, OR externaliz*, OR “disruptive behavior disorder”, OR DBD,) AND the gene of interest (“COMT” OR “Catechol-O-methyltransferase”) AND age of interest (youth OR child* OR adoles*). The key words were searched in the title and abstracts of online databases of PubMed, Embase, OVID Medline and PsychInfo, which gave 363 results in total. Results were restricted to articles published from 1947 to February 2021. After the duplicates were removed, 160 articles remained for further screening with the inclusion and exclusion criteria (Figure 1).

The data selection, collection and extraction processes were independently performed and confirmed by TK and EK. The inclusion criteria were studies that were experimentally designed, published in peer-reviewed journals, published in English, based on human participants and consisting of participants aged from 0 to 18 years old. Studies in languages other than English, case studies, studies with animals, studies that included participants older than 18 years old, studies with overlapping samples, studies that looked at the effects of multiple genes combined or a polymorphism different than Val158Met, or tested for outcomes other than endophenotypes of psychopathy were excluded. The review articles were excluded after a search in their reference lists, which did not provide additional eligible articles. The authors of conference abstracts were contacted for full texts which provided 2 additional articles for full-text review. Moreover, the authors of the articles in a language other than English were contacted for a possible English manuscript; however, this did not yield any additional articles.

### 2.2. New Evidence—Original Data

#### 2.2.1. Participants

The study participants were recruited for an ongoing study of childhood aggression at the Centre for Addiction and Mental Health (CAMH) in Toronto, Canada [62]. A total of 293, 142 male and 151 female, clinically aggressive and healthy community youth were included in this study (mean 12.41 (SD = 2.86) years). Inclusion and exclusion criteria have been described previously [62]. Briefly, youth with an intelligence quotient below 70, or with chronic medical illnesses, psychotic disorders, autism or Tourette’s Syndrome were excluded. Participants were of European ancestry, as determined by the principal component analysis of the genotyped variants using the HapMap CEU population as reference data. Part of the dataset has previously been analyzed in Hirata et al. [54]; however, the sample size has been expanded in the current study. Psychopathic traits were measured quantitatively from controls through clinically aggressive youth. The sample was split into two groups by the age of 13 years (above/equal to age 13 and below age 13), consistent with the previous literature [27,47,53], for the analyses of developmental subgroups separately.

Written, informed consent was collected from each participant and/or their guardians. This study was approved by the Research Ethics Board at the Centre for Addiction and Mental Health, Canada.

#### 2.2.2. Measures

Participants and their guardians were asked to fill out the Child Behavior Checklist (CBCL) and the Psychopathy Screening Device (PSD) [63], where they were asked to rate their children’s behavior within the last 6 months from Not True to Very True. The aggression, attention deficit, conduct, and oppositional defiant disorder subscales of CBCL, and the narcissism, impulsivity, and callous-unemotional subscales of PSD, were used to analyze the association of Val158Met with psychopathic traits.

#### 2.2.3. Genotyping

Genomic DNA was extracted from saliva, blood, and buccal cells. The Val158Met polymorphism of *COMT* (rs4680) was genotyped with the Illumina PsychArray Beadchip v.1.2 and v.1.3 (Illumina, San Diego, CA, USA), in CAMH Toronto, Canada. Quality control analyses were completed for the full genotype, where standard individual and SNP variant quality control procedures were performed using PLINK [64] and R [65] Biostatistics Softwares. Participants with missing genotypes >5%, sex-mismatch between genotyped and reported sex, abnormal heterozygosity and those who were related to another participant in this study were excluded. Departure from the Hardy–Weinberg equilibrium (HWE) was used for genotyping quality control. The rs4680 genotypes were extracted from the full genome-wide data, and genotype frequencies of Val158Met did not deviate significantly from HWE (*p* = 0.09).

#### 2.2.4. Analyses

Data analysis was performed using PLINK [64] and R [65] Biostatistics Softwares. To account for the non-normally distributed outcome variables, a series of Wilcoxon Rank-Sum Tests were conducted to analyze the CBCL and PSD subscale scores. Based on the previous results on the differential effects of the Val and Met alleles with development [50,66], statistical tests based on the Val dominance model were conducted using Met/Met versus Val allele carriers of *COMT* as the independent variable. Val carrier youth were coded as “0”, whereas youth with the minor allele Met/Met were coded as “1”. As different associations with the Val158Met genotypes were observed in females and males in previous studies, females and males were analyzed separately. Similar analyses were also conducted with a series of Kruskal–Wallis tests to analyze the association between the three genotypes (Val/Val vs. Val/Met vs. Met/Met) and the CBCL or PSD subscale scores. The results were further analyzed with Dunn’s Multiple Comparison Test.

A multiple testing correction for non-independent tests, M_eff_ [67], was used, where the effective number of tests were predicted to be 3.7, calculated from the correlations among the variables being tested. The corrected alpha level was calculated to be 0.014, from the overall alpha level divided by M_eff_: 0.05/3.7. 

## 3. Results

### 3.1. Search Results and Data Extraction

After the application of inclusion and exclusion criteria and full-text review, 28 studies that analyzed the effects of Val158Met on psychopathic traits in youth remained (Figure 1) and are summarized in Table 1. Data extraction included (i) the authors and the year of the article, (ii) sample size, (iii) age, (iv) behavioral disorder diagnoses when applicable, (v) ethnicity, (vi) sex, (vii) assessment tools and criteria, (viii) environmental factors when applicable and (ix) the key findings of the studies.

### 3.2. COMT in Psychopathic Traits—Systematic Review

Studies examining the role of Val158Met in psychopathic traits in youth have demonstrated mixed and conflicting results. As different studies used and analyzed different environmental factors, sex ratios, populations, and age groups, the reporting and comparing of the results will be divided with their subsequent categories. Our review/study will provide an opportunity for more in-depth analyses.

#### 3.2.1. Environmental Factors

Different environmental factors have been found to have significant effects on the association between Val158Met and psychopathy-related behaviors. Two of the most common environmental factors that have been studied in relation to *COMT* and psychopathy-related behaviors are socioeconomic status (SES) and serious life events. Hygen et al. [68] demonstrated that *COMT* did not have a significant main effect on aggression. On the other hand, the interaction between the *COMT* genotype and serious life events on aggression was significant, where children aged 4–5 years old with the Val/Val genotype were more susceptible to exhibiting aggressive behaviors after experiencing serious life events. Interestingly, Val/Val carriers displayed lower aggression scores than Met carriers without serious life events, demonstrating the importance of the gene–environment interaction on psychopathy-related behaviors [68]. The gene–environment interaction has also been studied by Nobile et al. [41], where they demonstrated that adolescents with the Val/Val genotype and low SES exhibit significantly higher scores on attention deficit/hyperactivity problems. On the other hand, a recent study by Abraham et al. [60] demonstrated that both boys and girls with at least one copy of the Met allele were more susceptible to being affected by early life adversities and express hyperactivity and impulsivity symptoms [60]. In both studies, Val158Met did not have a significant main effect on the hyperactivity and impulsivity scores.

The gene–behavior association may also be significantly affected by the early-caregiving environment and parental factors. A group of youth exhibited a significant interaction of the *COMT* genotype and parental separation on externalizing problems, where adolescents with the Met allele had the most externalizing problems among the children if they had separated parents. However, they did not exhibit significantly increased externalizing problems if their parents were together [69]. Similar to the results from the previous study, Zhang et al. [66] demonstrated that the Met allele carriers were more sensitive to the parenting styles. A sample of Chinese Han youth reported a similar association where children with at least one Met allele were more susceptible to exhibiting reactive aggression, but not proactive aggression, when they experienced low positive parenting. However, interestingly, the parenting style may also be influenced by the child’s genotype, which can further influence the gene–behavior association. A study with youth from the third, sixth and ninth grades showed that emotionally stable and extraverted parents demonstrated higher positive parenting to children with the Val/Val genotype than compared to the children with a Met allele. The difference in parenting may provide an explanation for why youth with the Val/Val genotype expressed decreased anger when compared with youth with the Met/Met genotype [70]. On the other hand, some studies have demonstrated that there is no significant interparental conflict and *COMT* interaction in predicting externalizing symptoms in youth aged 13 [71].

Another environmental factor that may affect the association between *COMT* psychopathy-related behaviors is prenatal maternal smoking. A study examining the interaction between maternal smoking during pregnancy and Val158Met on aggressive behaviors in adolescents found that while Val158Met alone did not significantly predict aggressive behaviors, youth with the Val/Val genotype and with mothers who smoked during pregnancy exhibited the highest rate of aggressive behaviors [72]. On the other hand, Salatino-Oliveira et al. [73] demonstrated that neither Val158Met alone nor in interaction with maternal smoking had significant effects on the conduct scores or crime rate in youth [73], similar to a previous study on youth conduct scores [74].

Low birth weight has also been associated with the exhibition of psychopathy-related behaviors in youth, making it an important environmental factor to study in relation to Val158Met. In a sample of youth with ADHD, Thapar et al. [74] found a significant interaction between Val158Met and low birth weight on conduct symptoms. Where children with the Val/Val genotype exhibited increased CD symptoms, children with both the Val/Val genotype and low birth weight had the highest conduct symptoms. On the other hand, two other studies that also studied mostly male youth with ADHD failed to replicate the results by Thapar et al. [74] and showed no interaction effect between Val158Met and birth weight on ADHD and CD symptoms [75,76]. 

Overall, when the environmental factors are being considered, there are mixed results regarding the risk genotype of the *COMT* gene. However, the significant interactions between the environmental factors and Val158Met on psychopathy-related behaviors signifies the importance of analyzing and considering the gene–environment interactions when studying the endophenotypes of psychopathy.

#### 3.2.2. Sex

Sex can significantly affect the results for the *COMT*–psychopathic traits association studies, which therefore needs to be considered carefully. Almost half of the studies (*k* = 13) that are included in this systematic review only examined male participants [77,78,79,80], or had a sex ratio that was significantly higher for males [28,46,75,76,81,82,83,84]. Two studies that only looked at males are reported below, the results of the remaining studies were explained in the other sections as they were also measuring the environmental factors or included samples with a diagnosis of a behavioral disorder.

A group of Russian male adolescent inmates who have been incarcerated for at least 6 months showed a significant positive association between the number of Val alleles and CD diagnosis and symptoms [80]. This high-risk sample also showed a significant association between Val158Met and ADHD symptoms, but with the Met/Met genotype [80]. Similarly, Nikolac Perkovic et al. [79] demonstrated that the Met/Met genotype was significantly less frequent in youth without hyperactive-impulsive and inattentive symptoms, where Met/Met carriers had higher hyperactive-impulsive and inattentive scores.

On the other hand, by studying a sample of both males and females, Amstadter and colleagues [84] demonstrated that only female youth, and not males, aged from 9 to 13 years old showed a significant correlation between the Met allele and increased risk-taking behaviors [84]. These results support that the interaction between Val158Met and psychopathic traits may be sex specific, and that males and females should be analyzed separately.

#### 3.2.3. Diagnosis of ADHD and/or Externalizing Disorders with Aggression

A factor that may significantly affect the *COMT*–psychopathic traits association is being diagnosed with ADHD or other externalizing disorders. As the literature indicates a possibility of the *COMT* genotype being a risk factor for psychopathic traits, there has been a high interest in understanding if it contributes to the externalizing disorders. Almost half of the studies (*k* = 12) included in this systematic review studied the *COMT*–psychopathic traits association in a specific population of youth with externalizing disorders such as ADHD, where the majority demonstrated a significant association between Val158Met and disruptive behaviors in children diagnosed with ADHD. One of the initial studies regarding Val158Met and psychopathic traits in youth with ADHD studied a sample of mostly male youth with clinical ADHD and found a significant association between the Val/Val genotype and increased CD symptom scores, when compared with the Met allele carriers [74]. These results were replicated by Caspi et al. [46], where they analyzed three different populations with and without ADHD, and found that the Val/Val genotype was significantly associated with increased CD symptoms, aggression and antisocial behaviors in the ADHD subgroup, however, the polymorphism was not a risk factor for behavioral problems in the control group [46]. They concluded that the Val158Met variant may be able to provide biological evidence to the subgroup of children within the ADHD cohort who would exhibit behavioral problems. Similar to the results from Caspi et al. [46], Salatino-Oliveira et al. [82] showed that the Val/Val genotype was significantly more frequent in children with disruptive behavior disorders such as ODD and CD comorbid with ADHD. Another study with mostly male participants with clinical ADHD showed similar results, where adolescents with ADHD who had the Val/Val genotype exhibited significantly higher emotional dysfunction psychopathy scores [28]. Moreover, a study in Chinese male youth with ADHD has also replicated these findings, where the authors showed a significant increased risk with the Val/Val genotype for ODD when compared with Met carriers, and the Val/Val genotype was more frequent in children who exhibited both ADHD and ODD [77]. Adding to the literature regarding the Val genotype and psychopathic traits in ADHD, children who are 7–8 years old also showed significant association between Val158Met and antisocial and CD symptoms in the presence of ADHD, where youth with the Val/Val genotype and ADHD had significantly increased antisocial behaviors [85]. Similar to the results from the previous studies, a sample of clinically-referred youth for externalizing disorders including ADHD, ODD and CD with a mean age of approximately 12 years old demonstrated that Val/Val carriers had significantly more commission errors in the A-X Continuous Performance Task than Met carriers, representing higher impulsivity. There was a significant interaction between Val158Met and clinical status on commission errors [86]. Moreover, another study confirming with the previous results found that, youth between 10–17 years of age, who were diagnosed with clinical ADHD or Hyperkinetic Disorder and had the Val allele showed significantly reduced response inhibition, fear empathy and autonomic responsiveness to the conditioned aversive stimulus. It was demonstrated that the difference in fear empathy and conditioning indirectly mediated the Val158Met–aggression link [87].

On the other hand, different studies demonstrated similar associations with the Met allele. A study with males diagnosed with ADHD demonstrated a significant association between Val158Met and impulsivity, where adolescents with the Met/Met genotype showed higher levels of impulsivity than Val allele carriers [78]. Another sample with youth from 6 to 13 years old with clinical ADHD replicated the results from the previous study, where the Met allele carriers had significantly higher scores on ADHD symptom severity, and carriers of the Met/Met genotype showed the highest rate of CD, with a non-significant trend [76]. Interestingly, in the sample that demonstrated a higher frequency of the Val/Val genotype in children with both ADHD and ODD, the Met allele was more frequent in children with ADHD only [77], raising an important question about the possibility of different genetic mechanisms behind ADHD comorbid with ODD.

Contrary to the previously reported results, there are also studies that demonstrated no significant association. One of the initial research studying youth with Hyperkinetic Disorder or ADHD demonstrated that there was not a significant association between Val158Met and neurocognitive performance including impulsiveness and response inhibition [81]. Similarly, Sengupta et al. [75] showed no significant effect of Val158Met on CD symptom scores in youth with ADHD. Another group of researchers also showed that Val158Met was not significantly associated with the CU scores in clinically-referred children for behavioral problems and aggression [54]. The results remained non-significant when the analyses were performed separately on child cases with and without ADHD diagnosis.

#### 3.2.4. Age

Age and development are additional factors that need to be recognized while analyzing and comparing the results of studies examining *COMT* and psychopathic traits. As the systematic review solely focused on youth, there were not many results that indicated change over time. However, a study showed that, even though the main effect of Val158Met on ADHD symptoms was not significant in early childhood, it became significant in middle childhood, where children with one copy of the Met allele had a higher risk of exhibiting ADHD symptoms than Val/Val carriers [60], indicating the importance of age and development in this area of research. 

As a summary, the results from the studies that included samples with behavioral disorder diagnoses have been mixed; however, the majority support the importance of Val158Met on the risk for psychopathic traits, with only three non-significant results. 

#### 3.2.5. Other Factors

##### Ethnicity

Ethnicity may also influence the association between Val158Met and psychopathic traits in youth, where African American participants show an increased chance of having the Val/Val genotype when compared with White participants [60]. Another study with a sample composed of 88% European Caucasian, and 22% African Canadian and mixed ancestry participants, only the European subgroup showed a significant association between Val158Met and CU scores, when the whole population did not [54]. These results support the importance of considering the ethnicity while recruiting and reporting the results.

##### Autism Spectrum Disorder and 22q11.2 Deletion Syndrome

Studies on the association between *COMT* and psychopathy-related behaviors were replicated in a sample with autism spectrum disorder and a sample with 22q11.2 deletion syndrome. Both studies showed significant associations between the *COMT* Val/Val genotype and increased scores on impulsivity and aggression [40,83].

### 3.3. Original Data

For male adolescents at or above age 13, Val allele carriers were associated with significantly increased CBCL aggressive scores (*p* = 0.03), impulsivity (*p* = 0.04), and callous-unemotional traits (*p* = 0.007). On the other hand, for male children below age 13, the Met/Met genotype was associated with increased conduct problems (*p* = 0.03) and impulsivity (*p* = 0.01). While female adolescents did not show any association between Val158Met and psychopathy-related behaviors, female children below age 13 with the Met/Met genotype had an increased risk for ADHD (Table 2). Results were also significant when three genotypes (Val/Val vs. Val/Met vs. Met/Met) were analyzed with Kruskal–Wallis test (Appendix A).

M_eff_ [67], a multiple testing correction for correlated tests, was used, where only the association with callous-unemotional traits for male adolescents at or above age 13, and the association with impulsivity for male children below age 13 survived the correction (*p* = 0.014). The results were not significant when the sample was analyzed as one group regardless of their age, and when females and males were further subgrouped based on their diagnosis of ADHD.

## 4. Discussion

### 4.1. COMT and Psychopathic Traits

The current systematic review summarized and reviewed the existing literature on the association between Val158Met and psychopathic traits in children and adolescents. Overall, there are mixed results in the literature; however, a few trends are observed within each group of analyses. First, environmental factors had a significant effect on the gene–behavior association. The majority of the studies showed that while Val158Met alone did not have a significant effect on the risk for children exhibiting putative precursors of adult psychopathy, youth with low SES [41], who experienced serious life events [68], had mothers who smoked while pregnant [72], or had low birth weight [74] exhibited significantly increased levels of psychopathy-related behaviors with the Val allele. On the other hand, youth with divorced parents [69] or a negative parenting environment [66] manifested risk from the Met allele. Moreover, studies have also shown that sex affected the association, where Met carriers had sex-specific effects [84]. For youth with a diagnosed behavioral disorder, while the majority of studies showed the increased risk with the Val/Val genotype [74], three studies demonstrated the risk with the Met allele carriers [76,77,78] and three studies did not have any significant associations. Furthermore, ethnicity may also play a role in the association given the fact that genotype distribution varies across different ethnicities [60], and a European subgroup showed a significant association while the whole population did not [54]. 

Development may also be a significant factor in the association between Val158Met and psychopathy-related behaviors, as demonstrated both in the literature and in the new evidence from our original data. Youth with the Met allele showed increased risk of disruptive behaviors in the 7–10 age range but not in earlier childhood years before age seven [60]. Results from our original data also provided the first evidence for the changing effects of Val158Met on psychopathic traits from childhood to adolescence. While children younger than age 13 exhibited increased risk with the Met/Met genotype, youth age 13 or older showed increased psychopathy-related behaviors with the Val allele. It is interesting to note that in the previous literature, several adult behavioral phenotypes have shown association to the Val carrier status, which is, in a general way, consistent with our Val association with post puberty age group [50,88]. These results suggest the importance of considering development when investigating the effects of genes on psychopathy-related behaviors, and may provide some explanation for the inconsistent findings in the literature.

### 4.2. COMT, Dopamine and Development

Increasing levels of COMT through development might provide an explanation for the differing contribution of Val158Met to children’s scores on putative precursors of adult psychopathy. Optimal dopamine levels in the PFC, mostly regulated by *COMT* enzyme levels, is shown to be crucial for decision making, controlling behavioral problems, antisocial behavior and substance abuse, where an increase in dopamine may lead to higher scores on psychopathy [89]. Studies have shown that *COMT* levels, thus enzyme activity, increase in the PFC with development since birth, leading to different enzyme levels in different age groups, independent of the genotype [47]. Therefore, the high activity effect of the Val allele may create more optimal levels in childhood, where it can compensate for the reduced COMT enzyme levels, while Met activity may not provide enough enzymatic activity, leading to increased dopamine amounts. On the other hand, the low activity of the Met variant may be optimal for balancing the amounts of dopamine in adolescence, as there will be less of a need from the genotype-dependent enzyme activity since COMT enzyme levels increase during development. This suggests that dopamine levels and behavioral control follow an inverse-U pattern, wherein the genotype required for the optimal level may differ with development [50,90]. As the influence of the COMT enzyme on dopamine levels depends on the enzyme activity levels, the functional role of Val158Met in the PFC may be age dependent, resulting in developmental subgroups having differing dopamine levels. However, the conflicting results in the literature regarding the increase in COMT activity, mRNA and protein levels with age and their association with Val158Met [47], emphasize the need for further studies examining the levels in relation to one another through development.

### 4.3. Clinical Implications

Studies have pointed out that psychopathic traits have high stability across the lifetime, and although they start in childhood, they can get worse over time [91,92] while getting less responsive to treatment for psychopathy [91,92,93]. Therefore, if effective detection, prevention and intervention techniques are applied for youth with high risk, the risk of ASPD and violent behaviors may be minimized [18,94]. One of the ways to develop early prevention and intervention techniques for psychopathy-related behaviors is to understand the etiology and risk factors. Psychopathic traits are found to be highly heritable [24], with a changing interaction with dopamine system and genes through development from childhood to adolescence [51]. Moreover, studies show that other risk factors, such as sex and different environmental factors, can interact and increase the risk in youth with a specific Val158Met genotype. Therefore, Val158Met, along with the consideration of developmental stage, may be useful as a prediction tool to identify genetically at-risk youth. It may also serve as a biomarker for the choice and dosage of medication leading to improved treatment efficacy for psychopathy in youth. Furthermore, it has potential to serve as a target for drug discovery and may possibly lead to new interventions and prevention. These novel strategies will benefit both the children struggling with behavioral disorders, and society by decreasing the future costs of criminal and violent behaviors [95] and the pain and suffering that commonly occur as the violent behaviors continue through adulthood.

### 4.4. Limitations

The current research has several strengths and limitations. One of the strengths of the systematic review is the systematic way of finding, including and excluding the articles, which makes the replication possible, while reducing the possibility of bias. Moreover, the inclusion of clinically aggressive sample and controls in the analysis of the original data may improve the analysis of the effects of gene on behavior by providing a clear distinction between high and low levels of psychopathic traits. Furthermore, a sample of both males and females in the original data allows for the analysis of sex-specific effects the polymorphism may have. However, there are still limitations that should be considered. First, in the systematic review, to limit the possibility of resulting in majorly unrelated articles, the search terms were only searched in the title and abstracts of the articles. However, this may have limited the access to the articles that tested the association between *COMT* polymorphism and endophenotypes of psychopathy, but did not mention the keywords in their titles or abstracts. Second, only articles in English were included due to language limitations of the author. Even though the authors of the articles in different languages were contacted for English manuscripts, there were no new articles provided with this method. This may have led to a possibility of underrepresentation of certain sample populations in the review, introducing a possible bias. Third, as the inclusion criteria included studies with participants until age 18 to exclude studies that were not focusing on youth, a few studies that examined adolescent groups with a range older than 18 were excluded. Even though this may have led to the exclusion of studies examining late adolescent groups, it was aimed to ensure that data from the early adult groups were not being compared with youth. Lastly, due to the main aim of the systematic review and inclusion-exclusion criteria, studies that examined *COMT* with different genes simultaneously with polygenic scores were excluded from this review. Limitations of the experimental study should also be considered. The relatively small sample size may have reduced the statistical power of our analyses, while the ethnically homogenous sample may have limited the generalizability of our results to European populations. Moreover, the current data did not account for the environmental factors that are likely to influence behavioral problems. Furthermore, recent research shows that gene-gene interactions in behavioral disorders are crucial, and a gene’s activation and function may be influenced by the polymorphisms of different genes and gene pathways [96,97]. As this systematic review and the novel study was based on a target gene alone, it may have not been able to capture the gene-gene interactions of *COMT* and their effects on endophenotypes of psychopathy.

### 4.5. Future Directions in the Field

With mixed results in the field, further research in the *COMT* gene is necessary to address the limitations of the current literature. First, from the studies that were included in the systematic review, the participants of the 8 of the studies were 100% Caucasian, while 12 of the studies were composed of more than 50% European or Caucasians. While only 2 studies were based on 100% of Chinese Han, only one study had the majority of the participants as African Americans with 56%. With limited generalizability of the results from the literature, and the ethnically homogenous sample of the original data suggests that future studies with larger sample sizes, diverse ethnicities are needed.

Another limitation that future research should focus on is the sex ratio and more studies with females. As mentioned before, almost half of the studies (*k* = 13) in this systematic review focused solely or majorly on males. Even though it may be an expected ratio for recruiting participants with behavioral problems [46], Amstadter et al. [84] demonstrated that only females had a significant association between the *COMT* genotype and risk-taking behaviors. Moreover, new evidence from the original data has further demonstrated a sex-specific effect of Val158Met on psychopathic traits. To make further conclusions about females, future research on females is required, as the results from males should not be generalized to the female adolescents. 

The conceptualization of the psychopathic traits in youth is a relatively new research area and these traits can be very heterogeneous among individuals. There are various methodologies that the researchers use to define the psychopathic traits [6], which leads to a possible concern about the summarization and comparison of the results from the literature. To improve the reliability and validity of results, future research should try to find a consensus on the definition and measurement techniques of psychopathic traits, such as using similar assessment tools or collecting measures from both the parents and teachers, as well as the youth.

Personality during childhood and adolescence fluctuates [6]. Even though the studies show high stability of psychopathic traits from childhood to adulthood [21,22,23], there has been limited focus on studying the endophenotypes of psychopathy and its association with the *COMT* genotype over time. The inclusion of different phenotypic expressions of psychopathy in youth in the original data provides an opportunity to measure the traits despite their possible change in expression through development. While we analyzed different age groups within the same study and design to test if the genetic association of Val158Met with these endophenotypes of psychopathy are stable over time, future research should also replicate the findings with longitudinal study designs.

Moreover, future efforts should focus on systematically reviewing the effects of Val158Met on psychopathic traits through development after early adolescence. Our current systematic review focused on youth until age 18, which therefore did not include studies from late adolescent and young adult participants. Even though the literature demonstrates that Val158Met continues to have effects on behavior in young adulthood, there is limited research with young adult participants. Results by Brennan et al. [72] indicate that individuals at age 20 with the Val/Val genotype, and with mothers who smoked during pregnancy, had increased levels of aggressive behavior. On the other hand, another study reported that young adults with the Val/Val genotype who were exposed to violence and experienced a positive parent–child relationship had decreased levels of aggressive behaviors [98]. More research is needed examining the association of Val158Met with psychopathic traits in young adults and systematically reviewing the literature with participants older than age 18 years.

Future research should also focus on the various components of the gene pathway as well as the possible epigenetic modifications. The difficulty of replicating the results of the role of *COMT* in psychopathic traits may also be due to the fact that *COMT* can interact with multiple genes and gene polymorphisms [96,97,99]. During the systematic search, we encountered only four articles studying the effects of multiple genes simultaneously (with *COMT*) on psychopathic traits with polygenic scores. As endophenotypes of psychopathy are likely polygenic, more research on the simultaneous effects of dopaminergic genes on psychopathic traits, with an emphasis on epistasis interactions, is necessary for discovering better prevention and treatment techniques. Moreover, studies have shown that environment has a significant effect on the *COMT*–psychopathic traits association [68]. These environmental factors can modify the genes by an epigenetic mechanism [100,101]. With limited focus on the epigenetic studies in the literature regarding the *COMT* gene and psychopathic traits, and no account for environmental and psychosocial factors in the presented new evidence, future research should examine the possible differing effects of environment and maltreatment with development on the gene–behavior association. Specifically, future research should examine the changes from the environmental factors in the molecular level, where they should study the effects of the epigenetic modifications in this gene–behavior association. 

Finally, future research should investigate the possible role of *COMT* as a biomarker for medication treatment response, particularly for drugs that influence the dopamine system [10,57]. Children versus adolescents may respond differently to treatment depending on their Val158Met genotype [57]. Therefore, *COMT* may have implications for the development of personalized medications for psychopathic traits, and may serve as a biomarker in adjusting the type and dosage of medications from childhood to adolescence. As drugs are commonly used for youth with disruptive behavior disorder, with over 70% being prescribed antipsychotics [102], future studies should examine the potential changing treatment response of youth to medications depending on their developmental stage and their Val158Met genotype.

## 5. Conclusions

To our knowledge, this article provides the first systematic review analyzing the role of *COMT* in childhood and adolescent psychopathic traits and the first empirical evidence that the effects of Val158Met on psychopathy-related behaviors may change through development. With inconsistent results in the literature, this review provides clearer information on the effects of Val158Met on endophenotypes of psychopathy, while emphasizing the importance of recognizing environmental and biological factors that may influence this association. Moreover, the results from the empirical study demonstrate the importance of age and development in the association between Val158Met and psychopathic traits, thus supporting that development is an additional factor that needs to be considered while analyzing these effects, and children and adolescents should be analyzed separately for accurate results. Overall, both studies emphasize the non-linear and non-static association between the *COMT* genotype and psychopathic traits. Learning about genetic risk factors while considering potential covariates, such as the developmental stage, for youth psychopathy can lead to specific early intervention and prevention strategies for different risk groups, thus reducing the risk for adult psychopathy from an early stage. While the majority of the results from the literature conclude that Val158Met has a significant association with the risk of psychopathy-related behaviors in youth, the conflicting results, along with the new evidence demonstrating the importance of development, emphasizes the need for further research. Future studies using similar methodologies and focusing on females and different ethnicities, on the developmental stability and the epigenetic modifications of this association, as well as analyzing multiple genes at the same time are necessary. Moreover, more studies are necessary to elucidate the possible role of *COMT* as a biomarker for predicting treatment response, while considering the influence of development on *COMT*–psychopathic traits association.

## Figures and Tables

**Figure 1 ijms-23-01782-f001:**
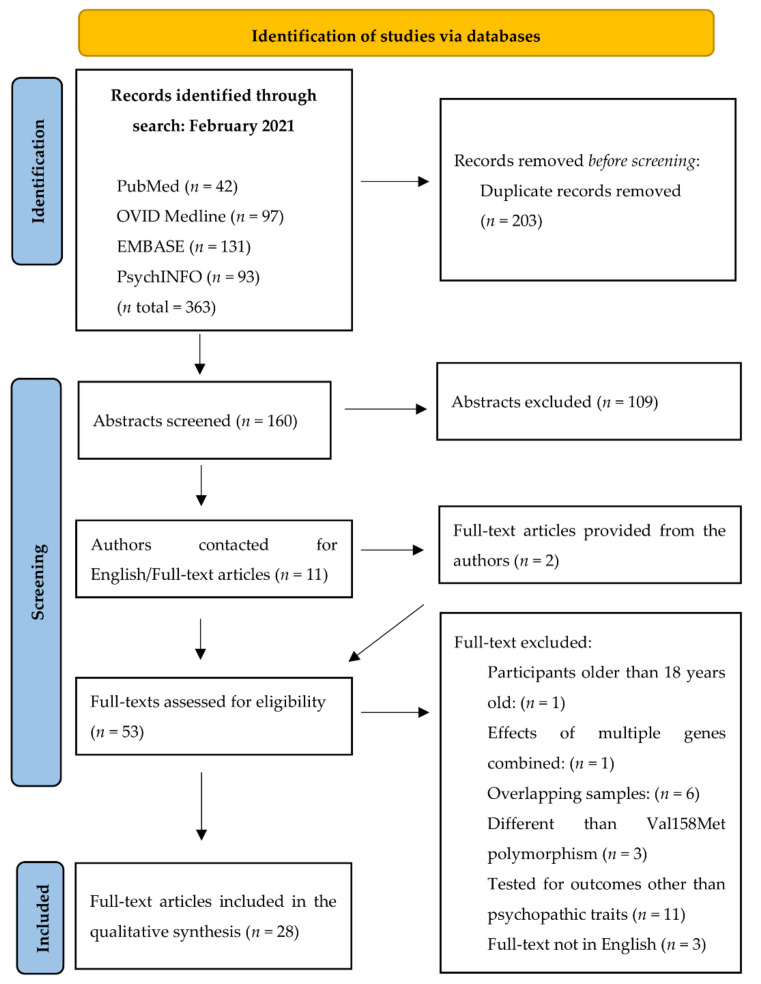
PRISMA flow diagram of the systematic review with inclusion and exclusion.

**Table 1 ijms-23-01782-t001:** The extracted data items from the studies analyzed in the systematic review.

Authors& Year of Publication	* N *	Age	Sample Characteristics: Clinical	Ethnicity	Sex F:	Sex M:	Behavioral AT	Environmental Factors	Environmental AT	Key Findings	Effect Estimates
Abraham et al. (2020)	1292	From birth to 11 years old	-	African American (56%) and White (44%)	50.1%	49.9%	Teachers: *DSM-IV* hyperactivity/impulsivity, and Inattention ADHD symptom severity	SES risk	Family income-to-needs ratio, household density, neighborhood safety, maternal education, a consistent partnership of a spouse/partner living in the home, maximum work hours of primary or secondary caregiver per week, and job prestige	Non-significant main effects of Val158Met and SES risk on hyperactivity/impulsivity in early and middle childhood. Significant Val158Met x SES risk on hyperactivity/impulsivity and inattention symptoms. Met carriers > Val/Val.	*p* = 0.108 (Val158Met—early) *p* = 0.323 (Val158Met—middle) *p* = 0.025 (Val158Met x SES—early) ⁺ *p* = 0.018 (Val158Met x SES—middle) ⁺
Davies et al. (2019)	279	From 13 to 16 years old	-	White (73%), African American (17%), multiracial (8%), other races (2%)	51.0%	49.0%	Parents: CBCL, Teachers: SDQ, Teacher’s Rating Scale of Child Actual Behavior, Adolescents: SDQ	Interparental conflict	Parents: interparental interaction task: verbal aggression, negativity and conflict, coerciveness, support, problem-solving communication, positive affect, negative escalation, cohesiveness Adolescents: security in the interparental subsystem	Non-significant interparental conflict x Val158Met in predicting externalizing symptoms.	*p* = 0.41
Salatino-Oliveira et al. (2016)	4095	From 11 to 15 to 18/19 years old	-	63.7% White	51.1%	48.9%	Age 11 and 15: SDQ for child and maternal mental health; age 18/19: self-report questionnaire about criminal behavior in the last 12 months	Prenatal maternal smoking	Perinatal assessment questionnaires	Non-significant effects of Val158Met on conduct scores and crime rate. Non-significant interaction between Val158Met *x* prenatal maternal smoking on conduct scores and crime rate *.	*p* = 0.932 (Val158Met—age 11) *p* = 0.472 (Val158Met—age 15) *p* = 0.282 (Val158Me *t*—age 18) *p* = 0.196 (Val158Met x smoking—age 18)
van Goozen et al. (2016)	194	10–17 years old (mean age: 13.95)	DSM-IV ADHD or ICD-10 Hyperkinetic Disorder	-	-	100.0%	DAWBA, SDQ, executive functioning: Wisconsin Card Sorting Task and Go/No-Go Task, cognitive and affective empathy through video-watching, fear conditioning using skin conduct response	-	-	Significantly reduced response inhibition, set shifting abilities, fear empathy and autonomic responsiveness to conditioned aversive stimulus in Val carriers < Met/Met.	*p* = 0.02 (Response inhib.) ⁺ *p* = 0.01 (shifting ability) ⁺⁺ *p* = 0.04 (fear empathy) ⁺ *p* = 0.001 (responsiveness to CS) ⁺⁺⁺
Zhang et al. (2016)	1399	12–13 years old (mean age: 12.32)	-	100% Chinese Han	47.2%	52.8%	PRQ	Maternal parenting	CRPR	Significant Val158Met x positive parenting on reactive aggression: Met carriers + positive parenting > Val/Val + positive parenting. Non-significant interaction between Val158Met x parenting on proactive aggression.	*p* < 0.01 (Val158Met x (+) parenting—reactive) ⁺⁺ *p* > 0.05 (Val158Met x (-) parenting—reactive) *p* > 0.05 (Val158Met x parenting—proactive)
Hygen et al. (2015)	704	4–5.58 years old (mean age: 4.56)	-	95.5% Norwegian	49.6%	50.4%	TRF	SLE	PAPA	Non-significant main effect of Val158Met on aggression. Significant Val158Met x SLE interaction on aggressive behaviors. Val/Val + SLE > Met carriers + SLE. Val/Val—SLE < Met carriers—SLE.	*p* = 0.78 (Val158Met) *p* = 0.02 (Val158Met x SLE (with SLE)) ⁺ *p* = 0.03 (Val158Met x SLE (without SLE)) ⁺
Park & Waldman (2014)	Clinically-referred: 224 Twin sample: 156	Mean age: 12.2	Clinically-referred for assessment of criteria for an externalizing disorder	80% European-American, 6% African American, 0.1% Hispanic 6.7% Other, %7.2 Missing	46.0%	53.9%	The A-X Continuous Performance Task	-	-	Significant main effect of Val158Met on commission error variability: Val/Val > Met carriers, representing impulsivity. Significant main effect of Val158Met on Signal Detection Theory indices (variability in sensitivity): Val carriers > Met/Met. Significant interaction Val158Met x clinical status on commission error variability.	*p* = 0.038 (Val158Met- commission errors) ⁺ *p* = 0.014 (Val158Met—SDT) ⁺ *p* = 0.010 (Val158Met x clinical status—commission errors) ⁺
Hirata et al. (2013)	144	6–16 years old (mean age: 10.8)	Clinically-referred for behavioral problems and persistent aggression & healthy adult controls	77.6% European Caucasian, 5.6% African Canadian, 16.7% mixed ancestry	27.8%	72.2%	CBCL, TRF, PSD	-	-	Non-significant effects of Val158Met polymorphism on callous-unemotional scores for the full sample. Significant association between Val158Met and CU scores in the European subgroup.	*p* = 0.173 (Val158Met—CU) *p* = 0.030 (Val158Met Europe—CU) ⁺
Oppenheimer et al. (2013)	263	9–15 years old (mean age: 12.03)	-	70% Caucasian, 7% African American, 6% Latino, 4% Asian/Pacific Islander, 13% other/mixed ethnicity	56.0%	44.0%	Videotaped psychosocial stressor challenge: Youth negative affect during stressor task	Parenting behaviors	Videotaped parent–child discussion: warmth and responsiveness Big Five Inventory for parent personality	Significant association between Val158Met and child anger: Met/Met > Val/Val. Emotionally stable and extraverted parents had higher positive parenting to children with Val/Val genotype.	*p* < 0.01 (Val158Met—anger) ⁺⁺ *p* = 0.02 (Val158Met—Parent warmth) ⁺
Nikolac Perkovic et al. (2013)	807	Median age: 10 and 15	-	100% Caucasian	-	100.0%	Teacher-report version of the SNAP-IV DSM-IV for ADHD	-	-	Significant difference between Val158Met genotype frequencies in youth with vs. without ADHD symptoms: Met/Met > Val carriers. Higher hyperactive-impulsive and inattentive scores in Met/Met > Val carriers.	*p* = 0.003 (Val158Met frequency) ⁺⁺ *p* = 0.008 (Val158Met—hyperactive-impulsive) ⁺⁺ *p* = 0.001 (Val158Met—inattentive) ⁺⁺⁺
Karam et al. (2013)	Case: 80 Control: 100	3–12 years old (mean age: 9)	ASD	100% Egyptian	17.0%	83.0%	CARS, CPRS-R:L	-	-	Significant association between Val158Met and hyperactivity scores: Val/Val > Met carriers.	*p* = 0.006; *p* = 0.03 (Val158Met—hyperactivity) ⁺
Amstadter et al. (2012)	223	9–13 years old (mean age: 11.3)	-	50% European American, 34.2% African American, 2.7% Latino, and 13.1% other (mixed ethnicity)	44.4%	55.6%	BART-Y	-	-	Significant correlation between Val158Met and BART performance in girls: Met carriers > Val/Val risk-taking behaviors.	*p* < 0.001 (Val158Met—BART in girls) ⁺⁺⁺ *p* = 0.47 (Val158Met—BART in boys)
Salatino-Oliveira et al. (2012)	516	Mean age: 10.55	ADHD	77.5% European-Brazilian	23.0%	77.0%	K-SADS-E, clinical evaluation of ADHD and comorbid conditions using DSM-IV criteria	-	-	Significant association between Val158Met and disruptive behavior disorders: Val/Val genotype was more frequent in children with ADHD comorbid with DBD (ODD and CD).	*p* = 0.016 ⁺
Nederhof et al. (2012)	1134	Mean age: 11.09 and 16.13	-	100% Dutch ancestry	52.0%	48.0%	YSR	Parental separation	Before age 11: TRAILS Family History Interview Between age 11 and 16: Event History Calendar	Significant Val158Met genotype x parental separation on externalizing problems: Met carriers + separated parents > Val/Val + separated parents. Met carriers + parents together ≈ Val/Val + parents together.	*p* = 0.03 (Val158Met x parents separate—externalize) ⁺ *p* = 0.29 (Val158Met x parents together—externalize)
Brennan et al. (2011)	470	15 and 20 years old	-	92% Caucasian ethnicity	57.0%	43.0%	Age 15 mother: CBCL, age 15 teacher: TRF, age 15 youth: YSR	Maternal smoking during pregnancy	Questionnaire (yes/no)	Significant Val158Met x maternal smoking during pregnancy interaction on aggressive behaviors at age 15: Val/Val + mothers who smoked >>. Val158Met genotype alone did not significantly predict aggressive behaviors. **	*p* < 0.05 (Val158Met x maternal smoking—aggression)⁺
Langley et al. (2010)	4365	7.5 years old (mean age: 7.65) and 8 years old (mean age: 8.59)	-	100% European origin	49.0%	51.0%	DAWBA, DSM-IV CD symptoms, Test of Everyday Attention for Children battery, Skuse Social Cognition Scale	-	-	Significant Val158Met x ADHD on antisocial behaviors (CD symptoms). Val/Val genotype + ADHD > Met carriers + ADHD.	*p* < 0.001 ⁺⁺⁺
Nobile et al. (2010)	618	10–14 years old (mean age: 12.1)	-	>95% Caucasian and of Italian ancestry	48.6%	51.4%	CBCL for DSM-oriented scales: ADHD problems, ODD, and CD	Socioeconomic status	Parental employment: Hollingshead 9-point scale	Significant Val158Met x SES on attention deficit/hyperactivity problems: Val/Val + low SES >> Non-significant effects of Val158Met genotype alone on behavior.	*p* = 0.004 (Val158Met x SES—ADHD) ⁺⁺ *p* = 0.420 (Val158Met—behavior)
Palmason et al. (2010)	166	6–13 years old (mean age: 9.7)	Clinically-referred for ADHD	-	15.7%	84.3%	Kinder-DIPS, DCL-HKS	Birth weight	Semi-structured, detailed interview with parents	Significant association between Val158Met and increased ADHD symptom severity: Met carriers > Val/Val. Non-significant Val158Met x birth weight effect on ADHD and CD.	*p* = 0.005 (Val158Met—ADHD) ⁺⁺ *p* = 0.697 (Val158Met x birth weight—ADHD/CD)
DeYoung et al. (2010)	174	Mean age: 16.23	Male adolescent inmates, incarcerated for at least 6 months	98% Russian ancestry	-	100.0%	K-SADS-PL	-	-	Significant association between Val158Met genotype and CD/ADHD. Diagnosis and symptoms of CD: Val/Val > Met carriers. ADHD symptoms: Met/Met > Val carriers.	*p* < 0.01 (Val158Met—CD diagnosis and symptoms) ⁺⁺ *p* < 0.05 (Val158Met—ADHD symptoms) ⁺
Paloyelis et al. (2010)	Case: 36 Control: 32	11–20 years old (mean age: 15.42)	ADHD	100% Caucasian	-	100.0%	Hypothetical delay discounting task, real-time delay discounting task, BIS-11A, Revised Conners’ Parent Rating Scales	-	-	Significant association between Val158Met genotype and impulsivity independent of ADHD diagnosis: Met/Met > Val carriers.	*p* < 0.05 ⁺
Albaugh et al. (2010)	149	6–18 years old (mean age: 10.93)	-	-	41.6%	58.4%	Mother rated CBCL: Aggressive Behavior scale & Attention Problems scale	-	-	Significant association between Val158Met genotype and aggressive behaviors (including direct and relational aggression): Met carriers > Val/Val. Non-significant association between Val/Val genotype and attention problems.	*p* = 0.016 (Val158Met—aggression) ⁺ *p* = 0.062 (Val158Met—attention)
Fowler et al. (2009)	147	12–19 years old (mean age: 14.5)	ADHD	100% UK White origin	7.5%	92.5%	Parent version CAPA, Child version of the CAPA, Child ADHD Teacher Telephone Interview, PCL-YV	-	-	Significant association between Val158Met genotype and emotional dysfunction psychopathy scores: Val/Val > Met carriers.	*p* = 0.02 ⁺
Qian et al. (2009)	171	6–17.5 years old (mean age: 10.3)	ADHD	100% Chinese Han	-	100.0%	Parents: CDIS Teachers: Rutter’s Scale	-	-	Significant association between Val158Met genotype frequencies and ADHD with co-morbid ODD: Val/Val more frequent in ADHD + ODD than ADHD alone (Met more frequent).	*p* = 0.019 (Val158Met—ADHD + ODD) ⁺
Caspi et al. (2008)	241 (a) 2232 (b) 1037 (c)	Clinical sample (a): 5–14 years old (mean age: 9.25) Birth cohort studies: 5 and 7 years old (b); 11.13 and 15 years old (c)	100% ADHD (a) 8% ADHD (b) 6% ADHD (c))	100% UK White origin 100% England and Wales 100% New Zealander	11% (a)	89% (a)	CAPA(a) Child ADHD Teacher Telephone Interview(a) mother and teacher report on criteria for ADHD specified by DSM-IV(b) CBCL(b) Diagnostic Interview Schedule for Children–Child Version(c) Adolescents followed to adulthood(c)			Significant association between Val158Met genotype and total number of CD symptoms, aggression and antisocial behavior: ADHD+ Val/Val > ADHD + Met carriers (a, b, c) ((a) is also reported in Thapar et al., 2005). Children without ADHD, there was no significant association between Val158Met genotype and aggression/antisocial behavior (b, c).	*p* = 0.05 (Val158Met + ADHD—CD symptoms) ⁺ *p* = 0.04 (Val158Met + ADHD—aggression) ⁺ *p* = 0.03 (Val158Met + ADHD—antisocial) ⁺ *p* = 0.38; 0.37 (Val158Met—aggression; antisocial (no ADHD))
Sengupta et al. (2006)	191	6–12 years old (mean age: 9)	ADHD	90.1% White, 4.2% Black, 1.6% Aboriginal, 3.6% Half-white, 5% Half-Asian	12.6%	87.4%	Parents: DISC-IV, Teachers: Conners Global Index-Teacher version questionnaire	Birth weight	Mother’s report	Non-significant main effects and interaction effect of Val158Met x birth weight on CD symptom scores.	*p* = 0.72 (Val158Met—CD) *p* = 0.71 (Val158Met x birth weight—CD)
Thapar et al. (2005)	240	5–14 years old (mean age: 9.25)	ADHD	100% UK White origin	11.3%	88.8%	CAPA–parent version: DSM-IV CD symptom score	Birth weight	Mother’s report	Significant main effect of Val158Met genotype and CD symptoms: Val/Val > Met carriers. Significant Val158Met x birth weight interaction on CD symptoms: Val/Val + low birth weight >>	*p* = 0.002 (Val158Met—CD) ⁺⁺ *p* = 0.006 (Val158Met x birth weight—CD) ⁺⁺
Bearden et al. (2005)	38	Mean age: 10.9	22q11.2 Deletion Syndrome	92% Caucasian ethnicity	61.0%	39.0%	CBCL	-	-	Significant association between Val158Met genotype and CBCL ratings (total/internalizing/externalizing problems scales, clinically behavioral problems): Val carriers > Met/Met.	*p* ≤ 0.01 = (Val158Met—total/ internalizing problems) ⁺⁺ *p* ≤ 0.05 = (Val158Met—externalizing problems, clinical behav. problems) ⁺
Mills et al. (2004)	124	6–16 years old (mean age: 9.2)	Meeting ICD-10 criteria for Hyperkinetic Disorder or DSM-III-R/IV criteria for ADHD	100% British Caucasian	8.0%	92.0%	Parents: CAPA, Teachers: CHATTI MFFT, CPT	-	-	Non-significant association between the Val158Met genotype and neurocognitive performance (impulsiveness and response inhibition).	*p* > 0.05

Notes: Assessment tools: AT; autism spectrum disorder: ASD; attention deficit hyperactivity disorder: ADHD; Balloon Analogue Risk Task—Youth Version: BART-Y; Barratt’s Impulsiveness Scale for Adolescents, version 11: BIS-11A; Child Behavior Checklist: CBCL; Childhood Autism Rating Scale: CARS; Child and adolescent psychiatric assessment: CAPA; Child ADHD Teacher Telephone Interview: CHATTI; Clinical Diagnostic Interviewing Scale: CDIS; Child Rearing Practices of Report: CRPR; Conduct Disorder: CD; Conners’ Parent Rating Scale Revised Long Version: CPRS-R:L; Continuous Performance Test: CPT; Diagnostic Interview Schedule for Children-IV: DISC-IV; Diagnostic and Statistical Manual of Mental Disorders, fourth edition: DSM-IV; Development and Well Being Assessment: DAWBA; German Hyperkinetic Syndrome diagnosis checklist: DCL-HKS; Schedule for Affective Disorders and Schizophrenia for School-Age Children, Epidemiological Version: K-SADS-E; Diagnostisches Interview bei psychischen Störungen im Kindesund Jugendalter: Kinder-DIPS; Schedule for Affective Disorders and Schizophrenia for School-Age Children: K-SADS-PL; Matching Familiar Figures Test: MFFT; Oppositional Defiant Disorder: ODD; Preschool Age Psychiatric Assessment: PAPA; Hare Psychopathy Checklist-Youth Version: PCL-YV; Psychopathy Screening Device: PSD; Proactive and Reactive Aggression Questionnaire: PRQ; Swanson, Nolan, and Pelham Questionnaire IV: SNAP-IV; Socioeconomic status: SES; Strengths and Difficulties Questionnaire: SDQ; Serious life events: SLE; Teacher’s Report Form: TR; Youth Self Report: YSR. ⁺ *p* < 0.05, ⁺⁺ *p* < 0.01, and ⁺⁺⁺ *p* < 0.001. * Crime rate during the age 18 or below are reported. ** Only included the results for age 15.

**Table 2 ijms-23-01782-t002:** Sample characteristics for males and females below and above age 13, based on the *COMT* Val158Met Val and Met alleles.

			Males		Females			
	Below Age 13 (*n* = 81)		Above Age 13 (*n* = 59)		Below Age 13 (*n* = 70)		Above Age 13 (*n* = 80)	
	Val Carriers (*n* = 58)	Met/Met (*n* = 23)	Val Carriers (*n* = 47)	Met/Met (*n* = 12)	Val Carriers (*n* = 52)	Met/Met (*n* = 18)	Val Carriers (*n* = 58)	Met/Met (*n* = 22)
Mean Age (S.D)	10.24 (1.95)	10.03 (2.28)	14.87 (1.58)	14.24 (1.04)	10.54 (1.82)	9.84 (2.12)	14.80 (1.26)	15.07 (1.45)
Mean CBCL Aggressive Behavior (S.D)	65.67 (16.16)	72.52 (16.18)	71.74 (13.83)	61.91 (12.31) *	60.9 (12.71)	61.63 (15.77)	71.12 (15.22)	66.45 (15.04)
Mean CBCL Attention Deficit / Hyperactivity (S.D)	59.07 (11.58)	63.77 (8.70)	64.32 (7.64)	58.43 (9.88)	57.91 (9.37)	59.5 (12.45) *	67.67 (9.87)	63.31 (11.53)
Mean CBCL Oppositional Defiant Problems (S.D)	61.32 (12.46)	65.92 (11.62)	67.77 (10.12)	61.13 (8.70)	58.67 (9.3)	61.2 (13.35)	67.75 (10.60)	66.23 (12.19)
Mean CBCL Conduct Problems (S.D)	61.46 (13.75)	71.07 (13.80) *	70.09 (11.98)	62.28 (10.96)	57.97 (10.97)	59.1 (12.44)	74.33 (13.53)	69.69 (13.78)
Mean PSD Narcissism Subscale (S.D)	0.60 (0.49)	0.81 (0.63)	0.94 (0.52)	0.64 (0.44)	0.47 (0.4)	0.31 (0.31)	0.89 (0.65)	0.63 (0.60)
Mean PSD Impulsivity Subscale (S.D)	0.87 (0.64)	1.25 (0.48) *	1.29 (0.58)	0.82 (0.67) *	0.68 (0.57)	0.77 (0.55)	1.22 (0.69)	1.03 (0.68)
Mean PSD Callous-Unemotional Traits Subscale (S.D)	0.89(0.42)	0.89(0.52)	1.09 (0.37)	0.78 (0.24) **	0.79 (0.37)	0.93 (0.43)	0.93 (0.49)	0.86 (0.43)

Note: * *p* < 0.05, ** *p* < 0.01.

## Data Availability

The data used to support the findings of this study are available from the corresponding author upon request.

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
