# Peer review of "COMT Val/Met and Psychopathic Traits in Children and Adolescents: A Systematic Review and New Evidence of a Developmental Trajectory toward Psychopathy"

_ijms, 2022, doi:10.3390/ijms23031782_

Round 1

Reviewer 1 Report

I think that the strongest key start point of this review is: “The literature suggests that focusing solely on CU traits and this subtype of CD while conceptualizing psychopathic traits in children may not be sufficient to detect a heterogeneous group of risk subjects [2,10]. Therefore, it has been suggested that including the combination of interpersonal grandiose-manipulative (narcissism, manipulative and egocentric, superficial charm) and behavioural daring-impulsive (impulsivity, novelty seeking, antisocial and irresponsible behaviours) traits along with the affective CU traits may provide a more comprehensive diagnostic measure for psychopathic traits in youth [2,10,11]”.

p.s. Please use American English (e.g behavioural and behaviours should be behavioral and behaviors)

The current review is interesting as it searched for association between COMT rs4680 SNP and early behavioral problems that may predict stability over time from childhood to adolescence and may even predict the development of adult psychopathy.

I have several concerns under a linguistic point of view. I find several sentences misleading and I suggest a deep revision of the review.

There is an overall consensus (even though some authors decide to use the term psychopathy with caution, e.g. Frick et al., 2003; da Silva et al., 2020; Colins et al., 2018) that it is not completely correct to use the term psychopathy when referring to youth, especially children. This is because psychopathy refers to a psychiatric disorder affecting adults. Nevertheless, it is now more and more clear that adult psychopathy follows a developmental trajectory and that some abnormal behavior preceding the onset of the disease is already present at early stages of life. This is the case of callous-unemotional traits, which resemble the interpersonal/affective deficiencies observed in psychopaths, and externalizing behaviors (e.g., conduct disorders and others) resembling the antisocial tendencies of psychopaths.

According to this theory, please revise the entire manuscripts, by removing the term psychopathy when describing youth, and either using CU traits or externalizing behavior when appropriate. Otherwise, you may choose to keep it you; in this case, one should explain why it is appropriate to say it discussing the articles mentioned above and others.

Consequently, also the title should be changed as it is misleading in the current version. It could be something like:

“COMT Val/Met and youth behavioral disorders: a Systematic Review and New Evidence of a Developmental trajectory paving the way for psychopathy”

Moreover, it would be important to give first a more detailed definition of adult psychopathy, report evidence of developmental trajectory, and then explain what psychopathic traits in youth are, in your opinion but based on the scientific literature such as the above-mentioned sentence. First, I think that it is important to keep in mind that psychopathic traits do not refer to single aspects of psychopathy (for example, only impulsivity or only lack of empathy), but combinations of both interpersonal/affective and antisocial behaviors. For example, I would not say that aggression is a psychopathic trait. Does it make sense? However, you could say that aggression is an endophenotype of psychopathy. So, please do not confuse “traits” with “endophenotypes”.

Introduction:

  • Line 88-89: “Variations in different target genes in the dopaminergic pathway have been found to be contributing to the risk of psychopathy” [17]. I do not have access to reference 17, however I do not believe this sentence as, to date, there is only one clear association between dopamine genetic variants and psychopathy traits: Fowler et al. 2009 found an association between COMT Val158Met and emotional dysfunction measure by PCL:YV.
  • Line 103: “The most common and well-studied single nucleotide polymorphism (SNP)”. I would say “The mosto commonly studied single nucleotide polymorphism (SNP)”.
  • Line 113: Catecholamine levels play a crucial role in the formation and expression of psycho-
  • 113 pathic traits [25]. I think that the reference number 25 is not appropriate here… there are not enough evidence in the literature to say that catecholamine are crucial for psychopathy. Are there? If so, please explain and provide references. Otherwise, one could just hypothesize it.
  • Line 119: “..children with one copy of the Met allele had increased risk for exhibiting psychopathic traits [27].” Here we go, please avoid the term psychopathic trait here. The article measured, by using a psychometric scale, specifically aggressive behavior. So, change psychopathic trait with aggressive behavior. Please use this approach throughout the paper.
  • Line 120: “However, there are inconsistent findings in the literature [28,29]..”. Here the two articles did not measure aggressive behavior specifically, they used CBCL showing associations with both internalizing and externalizing behavior, and no association, respectively, which is fine but please be precise and state exactly the precise results.
  • Line 126: “Developmental age is also another factor as the expression levels and effects of the gene as well as the expression of psychopathic traits may change over time especially during puberty, making it difficult to compare the two pre- and puberty age groups [32,33].” This is another key concept for the current review. On this matter, please provide literature data supporting this theory, in particular data suggesting a possible mechanism of action underlying this interesting process.

Statistical analysis:

  • Describe the methods used for post hoc correction in the statistical analysis section.
  • Line 266: “The aggression, attention deficit, conduct, and oppositional defiant disorders sub-scales of CBCL, and narcissism, impulsivity, and callous-unemotional sub-scales of PSD were used to analyze the association of Val158Met with psychopathic traits”.

Here, it is stated that the association of COMT Val158Met was tested with the following subscales: 1) aggression, 2) attention deficits, 3) conduct disorder, and 4) oppositional defiant disorders, 5) narcissism, 6) impulsivity, and 7) callous-unemotional. This makes 7 tests. According, for example, to the most conservative Bonferroni method, the alpha level should be set as 0.05/7, which makes 0.007. Please correct it. Report these details in the statistical analysis section.

Results

  • Line 341: “Replicating the previous findings, Nobile et al. [54]..”; actually, the finding has not been replicated by Nobliem as he investigated a different phenotype and a different negative environment; you may say “The same allele has been assiociated also with…
  • Line 356: “A sample of Chinese Han youth replicated the findings..”, same thing as before, this is not a replication study. Please adjust as suggested before.

Concerning the experimental study:

Please explain why you choose the age 13 as cut off.

Please report the limitations concerning the experimental study: like the small sample size.

Reviewer 2 Report

The systematic review presented by Kant et al is timely and well written. It aims to provide new evidence on the changing effects of COMT Val158Met on psychopathic traits in childhood and adolescence.

Several minor comments are as follows for consideration:

  • Although COMT Val158Met was the primary focus of this review, however, to begin with, other genotypes (such as rs6269 and rs4818, etc) of COMT associated with the risk of psychopathic traits should be adequately mentioned. Otherwise, the rationale setting up for the review could be weak if COMT Val158Met did not stand out.
  • Information of environmental factors should be included in the tables (Table 2 and Supplement Table 2.
  • Honestly, the final extracted data are from references with a wide year span (2004-2020). I suggest the author use the term “Systematic Review on the Evidence of Developmental Effects” in the title.
  • The aim of the present review is to provide a new insight into the expressed traits of psychopathy and its association with the COMT genotype over time. It is recommended that investigations regarding the effect of COMT Val158Met on psychopathic traits from young adulthood should be included, or at least thoroughly discussed.

Round 2

Reviewer 1 Report

The manuscript has been positively changed according to my comments.

Three additional comments:

  • About the databases used for the literature search: in the abstracts it is reported "Literature search was conducted using PubMed, EMBASE, 22 MEDLINE and PsychINFO". Pubmed is an interface used to search MEDLINE, thus y they are the basically the same thing. But then, in the material and methods, you have reported OVID MEDLINE...thus I think that you  have to correct it in the abstract.
  • paragraph of environmental factors (from line 386): in may opinion, here it is more appropriate to use "psychopathy-related behaviors" or "psychopathy endophenotypes" "instead of psychopathy traits".
  • - line 577:

    "COMT-psychopathic traits were replicated in a sample with autism spectrum disorder and a sample with 22q11.2 deletion syndrome. Both studies showed significant associations between COMT Val/Val genotype and increased scores on  impulsivity and aggression [83,88]. " Please change  "psychopathic traits" with "psychopathy-related behaviors" or with "psychopathy endophenotypes"
